# CCD-3DR: Consistent Conditioning in Diffusion for Single-Image 3D Reconstruction

## Abstract

In this paper, we present a novel shape reconstruction method leveraging a diffusion model to generate a 3D sparse point cloud for the object captured in a single RGB image. Recent methods typically guide a diffusion model with global shape information or local image features. However, such strategies fail to consistently align the denoised point cloud with the given image, leading to unstable conditioning and inferior performance. In this paper, we exploit a novel Centered Diffusion Probabilistic Model (CDPM) for consistent local feature conditioning. We constrain the noise and sampled point cloud from the diffusion model into a subspace where the point cloud center remains unchanged during both the forward and reverse diffusion process. Upon CDPM, we build CCD-3DR for single-image 3D reconstruction, where the stable point cloud center further serves as an anchor to align each point with its corresponding local projection-based features. Extensive experiments on synthetic benchmark ShapeNet-R2N2 demonstrate that CCD-3DR outperforms all competitors by a large margin, with over 40% improvement. We also provide results on the real-world dataset Pix3D to thoroughly demonstrate the potential of CCD-3DR in real-world applications. The code will be released soon.

## 1 Introduction

Single-image object reconstruction is a well-known ill-posed problem. While deep learning methods have made remarkable strides in achieving high-quality reconstruction, further improvements are still necessary to meet the demands of real-world applications (Zhai et al., 2023; Yang and Scherer, 2019). Recently, a new wave of methods leveraging **D**enoising **D**iffusion **P**robabilistic **M**odel (**DDPM**) (Ho et al., 2020) has emerged (Cheng et al., 2023; Melas-Kyriazi et al., 2023b; Luo and Hu, 2021; Melas-Kyriazi et al., 2023a; Poole et al., 2023), showcasing superior performance in various domains. For single-image 3D reconstruction with diffusion models, DMPGen (Luo and Hu, 2021) and PC$^2$ (Melas-Kyriazi et al., 2023b) are two representative baselines. In DMPGen, the condition is the global embedding of the target object, while in PC$^2$, in each step of the reverse process, the denoised point cloud is back-projected onto the feature map of the image to extract local feature for each point, which serves as the condition for the next reverse step.

However, directly applying diffusion models in single-image 3D reconstruction suffers from an inevitable challenge: uncontrollable center deviation of the point cloud, as shown in Fig. 1. (a). Since each point inside the point cloud and predicted noise is independently modeled, under the single-image reconstruction setting, no geometric or contextual priors can be harnessed to control the point cloud center. After each step of the reverse process in DDPM, the centroid of the generated point cloud will be shifted slightly. Therefore, from a random sampled Gaussian noise towards the target object, in the reverse process, the center of the point cloud will continuously undergo disturbances until it reaches the center of the target object. Based on our experimental findings, we have identified two problems caused by this center deviation.

**First**, the diffusion network needs to allocate capacity to handle the displacement of the point cloud center. It is crucial to ensure that the transition of the point cloud center from the initial Gaussian noise state to the final object reconstruction is appropriately managed. However, since the overall resource is limited, allocating network capacity to recover the center results in inferior performance in shape reconstruction. **Second**, the center deviation causes misalignment and inconsistency in the local feature conditioning, as used in PC$^2$ (Melas-Kyriazi et al., 2023b).

The misaligned feature adversely affects the subsequent denoising process in DDPM and degrades the overall quality of the final reconstruction. We explain more details of these two points in the Sec. 3.2.

To address the aforementioned problems, in this paper, we present a simple but effective method, CCD-3DR, which takes a single RGB image with the corresponding camera pose as input and reconstructs the target object with a sparse point cloud. Instead of directly leveraging the off-the-shelf DDPM, we propose a novel **C**entered **D**iffusion **P**robabilistic **M**odel (**CDPM**) that can enable consistent local feature conditioning in diffusion, which further significantly boosts the single-image reconstruction quality. Our core idea is to constrain the added noise in the diffusion process as well as the predicted noise and the sampled point cloud in the reverse process into a smaller subspace of the entire sampling space. With such constraints, CDPM sacrifices some of DDPM's generation diversity, yet it stables the point cloud center in exchange. In this subspace, the center of the corresponding noise of the point cloud coincides with the origin throughout the diffusion and reverse processes, as shown in Fig. 1. (b). Thereby, the point cloud center serves as an anchor in local feature extraction to align the point cloud with its corresponding projections consistently.

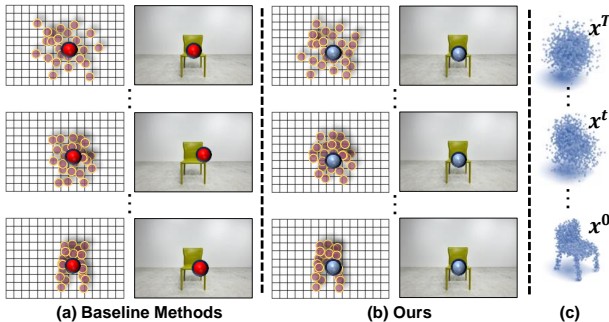

Figure 1: **Baseline Methods *vs* Ours in Conditioning.** In reverse process (c), the generated point cloud is back-projected onto the feature map of the RGB image, and local features are extracted around the projections. Fig (a) and (b) schematically compare the local feature extraction baseline (Melas-Kyriazi et al., 2023b) and Ours. In (a), during the reverse process, the point cloud center gradually deviates (indicated by red points). These deviations result in misaligned feature extraction, leading to a degradation in shape reconstruction quality. In contrast, our method, shown in (b), maintains the projection of the point cloud center unchanged throughout the reverse process (highlighted by blue points), serving as an anchor point that facilitates consistent extraction of local features.

Based on CDPM, we design CCD-3D for single-image 3D object reconstruction. In CCD-3D, to ensure that the noise and point cloud lie in the subspace defined in CDPM, a straightforward strategy is to iteratively generate samples in the entire space until one sample lies in the subspace. However, this is time-consuming and infeasible in real implementations. Instead, we first sample the noise in the entire space and centralize it. Next, we denoise the point cloud after predicting the noise using the diffusion network. In the subsequent process, these are then transferred to the subspace. We follow PC$^2$ (Melas-Kyriazi et al., 2023b) to back-project the point cloud onto the feature map of the image to extract local features around each projection.

In summary, our contributions are listed as follows, **(i)** We propose a novel centered denoising diffusion probabilistic model CDPM, which constrains the noise and point cloud in diffusion and reverse processes into a subspace where the point cloud center is forced to coincide with the origin. **(ii)** We present a new single-image 3D object reconstruction pipeline, CCD-3D, which leverages CDPM to consistently collect local features for the point cloud in diffusion, leading to superior performance in reconstruction quality. **(iii)** We evaluate CCD-3D on the synthetic dataset ShapeNet-R2N2 (Chang et al., 2015; Choy et al., 2016) to demonstrate its superiority over competitors. CCD-3D outperforms state-of-the-art methods by over $40\%$ under F-Score. Additional experiments on the real-world dataset Pix3D (Sun et al., 2018) demonstrate the potential of CCD-3D in real applications.

## 2    RELATED WORKS

3D reconstruction of the object shape from a single image has been a research focus in the community (Kar et al., 2017; Wang et al., 2018; Wu et al., 2017; Kar et al., 2015; Li et al., 2019; 2018; Zhang et al., 2021; Mao et al., 2021). Although it is an ill-posed problem, the shape priors of large-scale training datasets can guide the reconstruction process with generalization ability.

**Non-Generative Reconstruction Models.** Early methods use 2D encoders (Ronneberger et al., 2015; He et al., 2016; Simonyan and Zisserman, 2015) to encode features and use 3D decoders (Çiçek et al., 2016; Tran et al., 2015) to obtain shapes. The pioneering work such as 3D-

R2N2 (Choy et al., 2016) uses the occupancy grids as object shape representations and a following LSTM (Hochreiter and Schmidhuber, 1997) to fuse inputs from multiple views for prediction. The 2D features are extracted by a 2D CNN and projected to the 3D occupancy grids with a 3D deconvolutional neural network. LSM (Kar et al., 2017) reprojects 2D features into voxel grids and decodes shapes from these grids using a 3D convolutional GRU (Cho et al., 2014). Pix2Vox series (Xie et al., 2019; 2020) enjoy a serial architecture composed of a pretrained 2D CNN backbone and 3D transposed convolutional layers with multi-scale fusion for enhanced voxelization. Since the voxel representations are limited by the resolution of voxel size, point cloud and mesh-based shape representations are favored to get rid of the limitation (Hu et al., 2021; Wang et al., 2020; Zhang et al., 2018; Henderson and Ferrari, 2019; Erler et al., 2020; Mandikal and Babu, 2019; Gkioxari et al., 2019; Wen et al., 2019; Pan et al., 2019; Huang et al., 2023). More recent works utilizes implicit representations such as signed distance functions (Park et al., 2019; Xu et al., 2019), occupancy networks (Mescheder et al., 2018; Chen and Zhang, 2019) or neural radiance fields for object shape generation (Yu et al., 2020; Wang et al., 2021; Jang and de Agapito, 2021). Despite the different shape representations, the above methods are restricted to auto-encoder architecture and suffer limited performances in comparison to generative models.

**Generative Reconstruction Models.** Generative reconstruction models, in contrast to the routines mentioned above, estimate the shape distribution in a more explicit way to generate plausible shapes. For the first time to generate point clouds from single-view images, Fan et al. (Fan et al., 2017) build a point cloud generation network upon variational autoencoders (VAEs) (Kingma and Welling, 2014) to generate multiple plausible shapes. By incorporating both VAEs and generative adversarial networks (GANs) (Goodfellow et al., 2014), 3D-VAE-GAN (Wu et al., 2016) samples latent codes from a single-view image as the condition and outputs 3D shapes through 3D GAN generators. However, It heavily relies on class labels for reconstruction. 3D-aware GANs such as StyleSDF (Or-El et al., 2022) and Get3D (Gao et al., 2022) can simultaneously synthesize 2D images and 3D detailed meshes. However, these methods suffer from instabilities and mode collapse of GAN training.

Recently, diffusion models (Song and Ermon, 2019; 2020; Ho et al., 2020) exhibit advanced generation ability in such as text-to-image (Rombach et al., 2021), text-to-shape (Nichol et al., 2022) areas, enjoying more stable training phase and elegant mathematical explainability. Thereby, various point cloud based tasks take advantage of diffusion models to get results of higher quality. DMPGen (Luo and Hu, 2021) firstly applies the diffusion process in the point cloud generation task. LION (Zeng et al., 2022) further generalizes the point cloud in the hierarchical latent space with diffusion. Similarly, Lyu et al. (Lyu et al., 2022) utilize the point diffusion for shape completion. Point-Voxel Diffusion (Zhou et al., 2021) combines multiple representations in the diffusion process to generate stable results. To get the texture information for the point cloud, (Nichol et al., 2022) generates colored point clouds as the diffusion output for better visualization. Theoretically, such methodology can be readily leveraged into the single-view reconstruction task by regarding the RGB information as the condition (Poole et al., 2023; Melas-Kyriazi et al., 2023b). The recent method $PC^2$ (Melas-Kyriazi et al., 2023b) projects point clouds in the reverse diffusion process onto the image plane to query 2D features as shape and color conditions. Our new diffusion paradigm CDPM can be compatible with recent work, such as DMPGen and $PC^2$, while providing more accurate results.

## 3 METHOD

In the following sections, we outline our methodology. We start by providing a brief overview of point diffusion models, laying the groundwork for our approach. Subsequently, we explain the enhancements we have made to the traditional DDPM with the intention of augmenting its effectiveness in the realm of single-image reconstruction. These adaptations result in our innovative Centered Diffusion Probabilistic Model (CDPM). Lastly, we provide a comprehensive explanation of our single-image reconstruction pipeline CCD-3DR, which is constructed based on CDPM.

### 3.1 PRELIMINARIES: DIFFUSION MODELS

Diffusion denoising probabilistic models are a class of generative models inspired by non-equilibrium thermodynamics. It can iteratively move a set of Gaussian noise toward a uniform

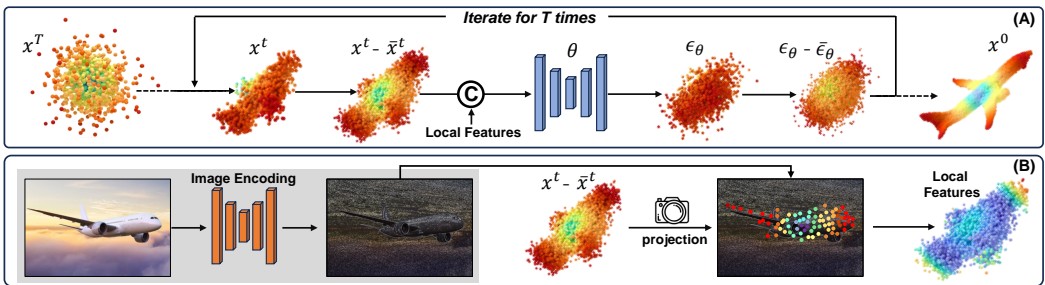

Figure 2: **Pipeline of CCD-3D.** Block (B) shows the local feature extraction process. Given a single RGB image (capturing the airplane) as the input, CCD-3D aims to reconstruct the object with CDPM. We first leverage a pre-trained MAE (He et al., 2022) model to extract feature maps from the image and interpolate them to the same size as the image (shown in the grey block). The feature maps provide local conditions for each point in the denoised centered point cloud $x^t - \bar{x}^t$ during the reverse process of CDPM. We back-project the centered point cloud onto the image and collect features around the projections to serve as the local features. Block (A) demonstrates the reverse process of CDPM. At step $t$, point cloud $x^t$ is first centralized to $x^t - \bar{x}^t$ and then concatenated with the local features out of Block (B). The U-Net denoiser $\theta$ predicts noise $\epsilon_\theta$ and centralizes it with $\epsilon_\theta - \bar{\epsilon}_\theta$. The point cloud $x^{t-1}$ can finally be recovered using Eq. 3.

and clean point cloud, capturing the target object. DDPM contains two Markov chains called the diffusion process and the reverse process. The two processes share a length of $T = 1K$ steps.

**Diffusion Process.** Let $p_0$ be the potential distribution of the complete object point cloud $x$ in the dataset and $p_T$ be the standard Gaussian distribution $p_T \sim \mathcal{N}(\mathbf{0}_{3N}, \mathbf{I}_{3N \times 3N})$, The diffusion process iteratively adds Gaussian noise $\epsilon$ into the clean data distribution $p_0$ according to the Markov Chain Rule until $p_0$ reaches $p_T$. Formally, let $x^0 \sim p_0$, then

$$q(x^{1:T}|x^0) = \prod_{t=1}^{T} q(x^t|x^{t-1}),$$

$$\text{where} \quad q(x^t|x^{t-1}) = \mathcal{N}(x^t; \sqrt{1-\beta_t}x^{t-1}, \beta_t \mathbf{I}). \tag{1}$$

The hyperparameter $\beta_t$ is pre-defined small constants. We use the subscript to denote the diffusion step $t$. Each $q(x^t|x^{t-1})$ is a Gaussian distribution and $q(x^t|x^0)$ can be reparameterized as,

$$q(x^t|x^0) = \sqrt{\bar{\alpha}_t}x^0 + \epsilon\sqrt{1-\bar{\alpha}_t}, \tag{2}$$

where $\alpha_t = 1 - \beta_t$, $\bar{\alpha}_t = \prod_{s=0}^{t} \alpha_s$, and $\epsilon \sim \mathcal{N}(\mathbf{0}, \mathbf{I})$.

From Eq. 2, for point diffusion, we can infer that if $x^0$ is sampled from a zero-mean distribution $p_0$, considering $\epsilon$ is also zero-mean, $q(x^t|x^0)$ can be modeled as a zero-mean distribution, which implies that for any $t \in [0, T]$, the diffusion process will generate a zero-mean distribution at this step. In this paper, we utilize this derivation to boost single-image 3D reconstruction.

**Reverse Chain.** The reverse process is also a Markov process that removes the noise added in the diffusion process. In this paper, the reverse process is conditioned on an RGB image $I$ capturing the object. We start with a sample $x^T \sim p_T$, and then iteratively sample from $q(x^{t-1}|x^t, f(I))$, where $f(I)$ denotes features extracted from $I$ to incorporate local or global supervision into the reverse process. When the number of sampling steps $T$ is sufficiently large, $q(x^{t-1}|x^t, f(I))$ can be well approximated with an isotropic Gaussian distribution with constant small covariance $\sigma_t^2$:

$$q(x^{t-1}|x^t, f(I)) = \mathcal{N}(x^{t-1}; \mu_\theta(x^t, f(I)), \sigma_t^2 \mathbf{I}),$$

$$\mu_\theta(x^t, f(I)) = \frac{1}{\sqrt{\alpha_t}}(x^t - \frac{\beta_t}{\sqrt{1-\bar{\alpha}_t}}\epsilon_\theta(x^t, f(I))), \tag{3}$$

where $\mu_\theta$ is the estimated mean. Thus, we can use the network parameterized by $\theta$ to directly learn $\epsilon_\theta$ under the condition $f(I)$.

**DDPM-Based Reconstruction** Consider a 3D point cloud with $N$ points, DDPM-based reconstruction methods (Luo and Hu, 2021; Melas-Kyriazi et al., 2023b) learn a diffusion model $S_\theta : \mathbb{R}^{3N} \to \mathbb{R}^{3N}$ to denoise the randomly sampled point cloud from $p_T$ into a recognizable object from target distribution $p_0$. Specifically, at each step $t$, the noise is predicted as the offset of each point from

| **Algorithm 1** CDPM: Training | **Algorithm 2** CDPM: Sampling |
|---|---|
| 1: **repeat** 
 2: $\quad x^0 \sim q(x^0), \quad x^0 = x^0 - \bar{x}^0$ 
 3: $\quad t \sim \text{Uniform}(\{1, 2, ..., T\})$ 
 4: $\quad \epsilon \sim \mathcal{N}(\mathbf{0}, \mathbf{I}), \epsilon = \epsilon - \bar{\epsilon}$ 
 5: $\quad$ Take gradient descent step on: 
 $\qquad \nabla_\theta \left\| \epsilon - \epsilon_\theta(x^t, f(I)) \right\|^2$ 
 6: **until** converged | 1: $x^T \sim \mathcal{N}(\mathbf{0}, \mathbf{I}), \quad x^T = x^T - \bar{x}^T$ 
 2: **for** $t = T, ..., 1$ **do** 
 3: $\quad \epsilon_\theta = \epsilon_\theta - \bar{\epsilon}_\theta$ 
 4: $\quad x^{t-1} \sim q(x^{t-1}\|x^t), x^{t-1} = x^{t-1} - \bar{x}^{t-1}$ 
 5: **end for** 
 6: **return** $x^0$ |

the current coordinate in $x^t$ to $x^{t-1} \sim q(x^{t-1}|x^t, f(I))$. Then we sample from $q(x^{t-1}|x^t, f(I))$ to obtain $x^{t-1}$. As for conditioning, DMPGen (Luo and Hu, 2021) encodes the given RGB image into a single global latent vector $z$ and concatenates $z$ with the obtained point cloud at each step during the reverse process. PC$^2$ (Melas-Kyriazi et al., 2023b) goes one step further by introducing local point-wise features for fine-grained geometry cues. It updates the local feature of each point at each step $t$ by back-projecting the point cloud $x^t$ onto the feature map using the known camera extrinsic $[\mathbf{R}_c|\mathbf{t}_c]$ and perspective projection matrix $\pi_c$,

$$Proj(x^t) = \pi_c(\mathbf{R}_c x^t + \mathbf{t}_c). \tag{4}$$

Then local features $f(I)$ around the projections $Proj(x^t)$ are aggregated with rasterization. These two methods (Luo and Hu, 2021; Melas-Kyriazi et al., 2023b) are selected as our baselines.

## 3.2 BOTTLENECKS IN DDPM-BASED RECONSTRUCTION

We now analyze the limitations of directly applying DDPM in 3D reconstruction like in DMPGen and PC$^2$ (Luo and Hu, 2021; Melas-Kyriazi et al., 2023b). Two bottlenecks are deteriorating the performance of these methods.

**First**, predicting the center bias is challenging for the network in the reverse process. Since we assume the variances are constant in all Gaussian distributions, we only need to analyze the center of each denoised point cloud. From $x^t$ to $x^{t-1}$, in Eq. 1 and 3, we have,

$$E(\bar{x}^{t-1}) = \frac{1}{\sqrt{\alpha_t}} E\left(\bar{x}^t\right), \quad E(\bar{\epsilon}_\theta\left(x^t, f(I)\right)) = 0. \tag{5}$$

Thus after sampling for $x^{t-1}$, we can obtain,

$$\bar{x}^{t-1} = \frac{1}{\sqrt{\alpha_t}} \left( \bar{x}^t - \frac{\beta_t}{\sqrt{1 - \bar{\alpha}_t}} \bar{\epsilon}_\theta\left(x^t, f\left(I\right)\right) \right) + \Delta_t, \tag{6}$$

where $\Delta_t$ is center bias generated by random sampling from Gaussian distribution for $x^{t-1}$. When $\bar{x}^T \neq \bar{x}^0$, the network $\theta$ needs to move the center of the denoised point cloud from $\bar{x}^T$ towards $\bar{x}^0$ under the following handicaps. First, $E(\bar{\epsilon}_\theta(x^t, f(I))) = 0$, while the network needs to predict non-zero-mean noise $\epsilon$ in several steps to move $\bar{x}^T \to \bar{x}^0$. Second, the network needs to overcome $\Delta_t$. Last, each point in $x^{T:0}$ is independently modeled in diffusion, and no constraints are incorporated to control the development of the point cloud center. Experiments in Sec. 4.1 demonstrate that accurately recovering $\bar{x}^0$ is a very hard job for the network. Wasting network capacity in the recovering center also results in poor performance in shape reconstruction.

**Second**, the change of the point cloud center makes the local feature conditioning inconsistent. As in PC$^2$, the difference $\Delta_{Proj}$ in projections of $Proj(\bar{x}^{t-1})$ and $Proj(\bar{x}^t)$ can be derived as

$$\Delta_{Proj} = \pi_c(\mathbf{R}_c(\bar{x}^{t-1} - \bar{x}^t) + \mathbf{t}_c). \tag{7}$$

If $\Delta_{Proj}$ is sufficiently large, the features collected for the point center can be totally different from $x^t$ to $x^{t-1}$, which will mislead the following denoising steps. Moreover, since we only use a single RGB image as a conditioner, we have no contextual or geometric constraints to rectify this misalignment.

### 3.3 FROM DDPM TO CDPM

To address the aforementioned bottlenecks, we propose a novel CDPM model designed for single-view 3D reconstruction. The core idea of CDPM is simple and straightforward yet effective. To eliminate the influence of center bias in the reverse process, we add the following constraint,

$$\bar{x}^t = \mathbf{0}, \quad t = 0, 1, 2..., T. \tag{8}$$

This constraint enforces the denoised point cloud in each step to be zero-mean so that the center remains unchanged during the reverse process. As shown in Eq. 2 and Eq. 3, if Eq. 8 holds, we have $\bar{\epsilon} = \mathbf{0}$, $\bar{\epsilon}_\theta(x^t, f(I)) = \mathbf{0}$. Let $\mathbb{S}_{x^t}$ denote the space of all possible samplings from the distribution $q(x^t | x^{t+1})$, then the space $\mathbb{S}_{x^t, \bar{x}^t = \mathbf{0}}$ under the constraint Eq. 8 is a subspace, *i.e.* $\mathbb{S}_{x^t, \bar{x}^t = \mathbf{0}} \subset \mathbb{S}_{x^t}$. Similarly, we define $\mathbb{S}_\epsilon$, $\mathbb{S}_{\epsilon, \bar{\epsilon} = \mathbf{0}}$, $\mathbb{S}_{\epsilon_\theta}$, $\mathbb{S}_{\epsilon_\theta, \bar{\epsilon}_\theta = \mathbf{0}}$. In summary, from DDPM to CDPM, we constrain $x^t, \epsilon, \epsilon_\theta$ all in a smaller subspace,

$$\begin{aligned} \text{DDPM} : & x^t \in \mathbb{S}_{x^t}, \epsilon \in \mathbb{S}_\epsilon, \epsilon_\theta \in \mathbb{S}_{\epsilon_\theta} \implies \\ \text{CDPM} : & x^t \in \mathbb{S}_{x^t, \bar{x}^t = \mathbf{0}}, \epsilon \in \mathbb{S}_{\epsilon, \bar{\epsilon} = \mathbf{0}}, \epsilon_\theta \in \mathbb{S}_{\epsilon_\theta, \bar{\epsilon}_\theta = \mathbf{0}}. \end{aligned} \tag{9}$$

Therefore, we prioritize the stability of the point cloud center to a certain extent, sacrificing a portion of the diversity in diffusion models. For point cloud $x^t$ in the reverse process, after obtaining $q(x^t | x^{t+1})$, we can sample multiple times until the sampled point cloud lies in $\mathbb{S}_{x^t, \bar{x}^t = \mathbf{0}}$. However, such a strategy is infeasible in real implementation. Thereby we simply first sample in $\mathbb{S}_{x^t}$ and then centralize the point cloud to project it into $\mathbb{S}_{x^t, \bar{x}^t = \mathbf{0}}$. The same holds true for $\epsilon$ and $\epsilon_\theta$.

Specifically, as explained in Alg. 1 and Alg. 2, we first build a dataset composed of $M$ data pairs $\mathcal{D} = \{(x_i, I_i) | 1 \le i \le M\}$, where $x_i$ denotes the $i$-th ground truth point cloud sampled from the object mesh, and $I_i$ is the corresponding RGB image capturing the object. Compared to DDPM, CDPM mainly makes improvements in three points:

**First**, the point clouds in $\mathcal{D}$ are centralized as $x_i - \bar{x}_i$, where $\bar{x}_i$ denotes the centroid of $x_i$, establishing a new zero-mean dataset $\bar{\mathcal{D}} = (\bar{x}_i, I_i)$.

**Second**, for noise $\epsilon$ added in the diffusion process for training and the noise $\epsilon_\theta$ predicted in the reverse process, we also centralize them with $\epsilon - \bar{\epsilon}$ and $\epsilon_\theta - \bar{\epsilon}_\theta$, where $\bar{\epsilon}$ and $\bar{\epsilon}_\theta$ denote the corresponding gravity centers.

**Third**, during inference, for $x^{t-1}$ sampled from $q(x^{t-1} | x^t, f(I))$, we also centralize it with $x^{t-1} - \bar{x}^{t-1}$. From Eq. 2, since we keep $x^0$ and $\epsilon$ to be zero-mean, the diffused point cloud in each step $t$ should be zero-mean.

The advantages of CDPM over DDPM in single-image reconstruction can be summarized as follows:

**First**, our reverse process starts with a zero-mean Gaussian noise and arrives at the zero-mean reconstruction $x^0$ after $T$-step zero-mean denoising. This zero-mean nature of the reverse process provides a useful regularization for the network to focus more on the shape of the object rather than tracking the center of the point cloud. Therefore, our CDPM outperforms the previous DDPM-reconstruction methods even with only global embedding of the object, like in (Luo and Hu, 2021).

**Second**, CDPM enables consistent local feature conditioning in the reverse diffusion process. As in PC$^2$ (Melas-Kyriazi et al., 2023b), the point cloud is back-projected onto the image feature map to extract local point-wise features as conditioning. However, due to the uncontrollable center bias in the reverse process, the projection of each point may gradually deviate, making the local feature aggregation process fail and further deteriorating the final reconstruction quality. In contrast to DDPM-based PC$^2$, our method CDPM keeps the centroid of the denoised point cloud in each step to coincide with the origin, which further serves as an anchor point in local feature collection. The projection of this anchor point remains the same in the reverse process and thus aligns the point cloud with the feature map to obtain consistent features.

### 3.4 CCD-3DR

For a fair comparison with baseline methods, we follow PC$^2$ (Melas-Kyriazi et al., 2023b) to use MAE (He et al., 2022) to extract 2D feature maps from the given RGB image. The feature maps are of equal length and width of the input image to facilitate point cloud projection. For the diffusion

network $\theta$ used to predict the noise $\epsilon_\theta$, we adopt the Point-Voxel CNN (PVCNN) (Liu et al., 2019). We use the classic $\mathcal{L}_2$ loss to supervise the training of $\theta$, as specified in Alg. 1.

## 4 EXPERIMENTS

**Datasets**. We evaluate CCD-3DR on the synthetic dataset ShapeNet-R2N2 (Choy et al., 2016; Chang et al., 2015) and real-world dataset Pix3D (Sun et al., 2018). ShapeNet contains a diverse collection of 3D models spanning various object categories, such as furniture, vehicles, and more. The dataset is meticulously annotated, providing not only the 3D geometry of the objects but also rich semantic information, making it an essential tool for the quantitative evaluation of single-view reconstruction methods. We follow baseline methods (Melas-Kyriazi et al., 2023b; Yagubbayli et al., 2021; Xie et al., 2020) to use the R2N2 (Choy et al., 2016) subset along with the official image renderings, train-test splits, camera intrinsic and extrinsic matrices. The R2N2 subset covers 13 categories in total. Pix3D (Sun et al., 2018) is a large-scale benchmark of diverse image-shape pairs with pixel-level 2D-3D alignment. Previous methods (Cheng et al., 2023; Xie et al., 2019; 2020; Sun et al., 2018) only harness the *chair* category and exclude the occluded samples. Since our method needs to use all data to demonstrate robustness towards occlusion, we leverage 3 categories: {*chair*, *table*, *sofa*} and randomly generate train-test split with about $90\%$ samples as the training set and the remaining as the testing set. Details are provided in the Supplementary Material.

**Implementation Details.** We implement CCD-3DR in PyTorch and evaluate the method on a single GeForce RTX 3090Ti GPU with 24GB memory. For ShapeNet-R2N2 (Choy et al., 2016; Chang et al., 2015), we first resize the provided images of size $137 \times 137$ to $224 \times 224$ and adjust the focal length accordingly. We follow prior work to use 8192 points in training and inference for fairness in computing the F-Score. On Pix3D (Sun et al., 2018), since the images are of different sizes, we first crop the image with the given bounding box to obtain an object-centric image and then resize it to $224 \times 224$. The camera intrinsic matrix is also adjusted correspondingly. During training, we train CCD-3DR with batch size 16 for 100K steps in total, following PC$^2$ (Melas-Kyriazi et al., 2023b). We use the AdamW optimizer with a dynamic learning rate with warmup which increases from $1 \times 10^{-5}$ to $1 \times 10^{-3}$ in the first 2K steps and then decays exponentially until 0 in the following 98K steps.

**Baselines.** We select DDPM-based DMPGen (Luo and Hu, 2021) and PC$^2$ (Melas-Kyriazi et al., 2023b) as our baseline methods. On ShapeNet-R2N2, we compare with the official results of PC$^2$. Since DMPGen doesn't provide results of single-view reconstruction on ShapeNet-R2N2, we reimplement it by using pre-trained MAE (He et al., 2022) to extract global shape code and then follow the diffusion process in the original paper to reconstruct the object, denoted as DMPGen$^*$. We provide three variants of CCD-3DR on ShapeNet-R2N2, in which *Ours* uses only local features like in PC$^2$, *Ours-G* leverages only global features as DMPGen$^*$ and *Ours-(G+L)* incorporates both local and global features for reconstruction, as shown in Tab. 4. On Pix3D, we retrain PC$^2$ and DMPGen$^*$ under the same settings of CCD-3DR.

**Evaluation Metrics**. We use Chamfer Distance (CD) and F-Score@0.01 following (Melas-Kyriazi et al., 2023b; Cheng et al., 2023) as the evaluation metrics. CD quantifies the dissimilarity between two sets of points by measuring the minimum distance from each point in one set to its nearest point in the other set. To compensate for the problem that CD can be sensitive to outliers, we also report F-Score with the threshold 0.01, *i.e.*, for each reconstructed point. If its nearest distance to the ground truth point cloud lies below the threshold, it is considered correctly predicted. Note that previous methods (Choy et al., 2016; Yagubbayli et al., 2021; Xie et al., 2020) typically report the results using the voxelized $32^3$ volume as the shape representation, which quantizes the sampled points and fails to reflect the reconstruction quality of fine-grained structures. Therefore, we follow PC$^2$ (Melas-Kyriazi et al., 2023b) to use sampled points from the object mesh as the ground truth. Results of other methods (Choy et al., 2016; Yagubbayli et al., 2021; Xie et al., 2020) are re-evaluated using the same setting for fair comparisons.

### 4.1 COMPARISONS WITH STATE-OF-THE-ART METHODS.

**Performance on Synthetic Dataset ShapeNet-R2N2.** In Tab. 1, we compare CCD-3DR with state-of-the-art competitors on ShapeNet-R2N2 under the F-Score@0.01 metric. 3D-R2N2 (Choy et al.,

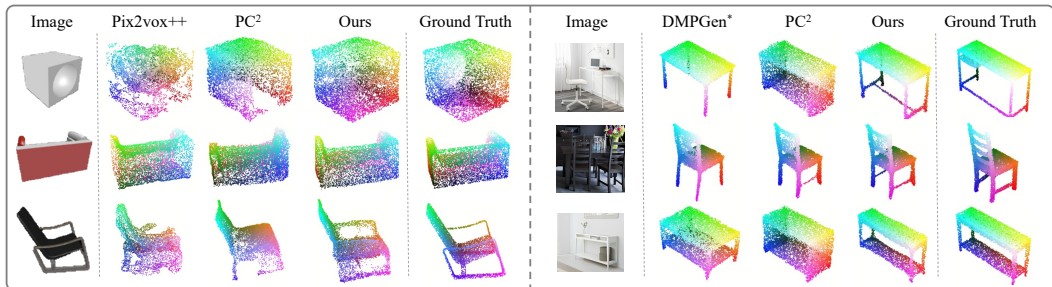

Figure 3: **Qualitative comparisons on synthetic dataset ShapeNet-R2N2 (Choy et al., 2016; Chang et al., 2015) (left) and real-world dataset Pix3D (Sun et al., 2018) (right).** Our method can recover fine-grained structures accurately, like the handle of the chair.

| Category | 3D-R2N2 | LegoFormer | Pix2vox++ | DMPGen* | PC$^2$ | Ours | DMPGen*(O) | PC$^2$(O) | Ours(O) |
|---|---|---|---|---|---|---|---|---|---|
| airplane | 0.225 | 0.215 | 0.266 | 0.454 | 0.473 | **0.725** | 0.565 | 0.681 | **0.785** |
| bench | 0.198 | 0.241 | 0.266 | 0.175 | 0.305 | **0.480** | 0.289 | 0.444 | **0.573** |
| cabinet | 0.256 | 0.308 | **0.317** | 0.087 | 0.203 | 0.282 | 0.111 | 0.303 | **0.371** |
| car | 0.211 | 0.220 | 0.268 | 0.310 | 0.359 | **0.395** | 0.402 | 0.420 | **0.466** |
| chair | 0.194 | 0.217 | 0.246 | 0.171 | 0.290 | **0.335** | 0.312 | 0.377 | **0.406** |
| display | 0.196 | 0.261 | 0.279 | 0.211 | 0.232 | **0.381** | 0.236 | 0.357 | **0.487** |
| lamp | 0.186 | 0.220 | 0.242 | 0.207 | 0.300 | **0.438** | 0.347 | 0.399 | **0.490** |
| loudspeaker | 0.229 | 0.286 | **0.297** | 0.113 | 0.204 | 0.219 | 0.126 | 0.288 | **0.291** |
| rifle | 0.356 | 0.364 | 0.410 | 0.474 | 0.522 | **0.762** | 0.663 | 0.686 | **0.828** |
| sofa | 0.208 | 0.260 | 0.277 | 0.078 | 0.205 | **0.293** | 0.106 | 0.298 | **0.349** |
| table | 0.263 | 0.305 | 0.327 | 0.155 | 0.270 | **0.427** | 0.310 | 0.420 | **0.488** |
| telephone | 0.407 | 0.575 | **0.582** | 0.333 | 0.331 | 0.423 | 0.464 | 0.523 | **0.598** |
| watercraft | 0.240 | 0.283 | 0.316 | 0.201 | 0.324 | **0.475** | 0.399 | 0.424 | **0.610** |
| Average | 0.244 | 0.289 | 0.315 | 0.228 | 0.309 | **0.433** | 0.333 | 0.432 | **0.519** |

Table 1: **Performance on ShapeNet-R2N2.** We compare our method with competitors under F-Score@0.01. The Oracle setting (marked as (O)) refers to predicting 5 samples of each image and selecting the best prediction as the final result.

2016), Legoformer (Yagubbayli et al., 2021), Pix2vox++ (Xie et al., 2020) are voxel-based methods, while DMPGen (Luo and Hu, 2021), PC$^2$ (Melas-Kyriazi et al., 2023b) are diffusion-based methods, serving as baselines of CCD-3DR. From Tab. 1, it can be clearly deduced that our method CCD-3DR achieves state-of-the-art performance in 10 out of 13 categories. Considering the *Average* performance, CCD-3DR outperforms previous best method Pix2vox++ with 0.433 *vs.* 0.315, about a 37.5% leap forward. Furthermore, compared with diffusion-based baseline method PC$^2$, CCD-3DR demonstrates superior performance under all the categories and improves PC$^2$ by 40.1%, with 0.433 *vs.* 0.309. We also report the **Oracle** results, following the setting in PC$^2$, where for each test image, we predict 5 possible reconstruction results and select the one with the highest F-Score@0.01 as the final result. Under the **Oracle** setting, our method surpasses all competitors by a large margin, with about a 20.1% improvement over PC$^2$ Oracle.

**Performance on Real-World Dataset Pix3D.** In Tab. 2, we compare CCD-3DR with other DDPM-based reconstruction methods using Chamfer Distance and F-Score@0.01. Our method consistently outperforms competitors in all categories. On average, CCD-3DR surpasses the second-best method PC$^2$ by 20% on ShapeNet-R2N2 and 15% on Pix3D.

**Qualitative Comparisons.** We provide visualization comparisons with previous methods in Fig. 3. It can be seen clearly that our method surpasses competitors with respect to the reconstruction quality. Particularly, due to our consistent feature conditioning scheme, our method showcases superiority in recovering fine-grained structures, like the hand of the chair. We provide more results in the Supplementary Material.

### 4.2 ABLATION STUDIES

We conduct several ablation studies on public datasets. Note that except for ablated terms, we leave all other terms and settings unchanged.

| Method | Chair | Table | Sofa | Average | Chair | Table | Sofa | Average |
|---|---|---|---|---|---|---|---|---|
| DMPGen* | 0.188 | 0.176 | 0.243 | 0.202 | 53.30 | 50.56 | 21.04 | 41.63 |
| PC$^2$ | 0.336 | 0.294 | 0.377 | 0.336 | 33.21 | 13.13 | 3.760 | 16.70 |
| Ours | **0.439** | **0.559** | **0.489** | **0.496** | **14.98** | **1.475** | **0.712** | **5.722** |

Table 2: **Performance on Pix3d.** F-Score@0.01 (left) and Chamfer Distance ($\times 10^{-3}$) (right) is reported in the table. Our method outperforms diffusion-based competitors.

| Occ. Ratio | Method | Chair | Table | Sofa |
|---|---|---|---|---|
| $\sim 20\%$ | PC$^2$ (Melas-Kyriazi et al., 2023b) | 0.324 | 0.280 | 0.365 |
| | Ours | **0.424** | **0.535** | **0.421** |
| $\sim 50\%$ | PC$^2$ (Melas-Kyriazi et al., 2023b) | 0.310 | 0.260 | 0.337 |
| | Ours | **0.411** | **0.520** | **0.397** |

Table 3: **Ablation studies of robustness towards occlusions.** Occ. Ratio refers to occlusion ratio. We report the F-Score@0.01 after randomly masking about 20% and 50% visible pixels of the image.

| Category | air-plane | bench | cabinet | car | chair | display | lamp | loud-speaker | rifle | sofa | table | tele-phone | water-craft | Average |
|---|---|---|---|---|---|---|---|---|---|---|---|---|---|---|
| Ours-G | 0.599 | 0.298 | 0.204 | 0.251 | 0.283 | 0.223 | 0.316 | 0.177 | 0.653 | 0.201 | 0.266 | 0.355 | 0.311 | 0.318 |
| Ours-(G+L) | 0.727 | 0.463 | 0.277 | 0.398 | 0.341 | 0.366 | 0.429 | 0.214 | 0.777 | 0.287 | 0.433 | 0.414 | 0.469 | 0.430 |
| Ours | 0.725 | 0.480 | 0.282 | 0.395 | 0.335 | 0.381 | 0.438 | 0.219 | 0.762 | 0.293 | 0.427 | 0.423 | 0.475 | 0.433 |

Table 4: **Ablations on the effect of local and global features on ShapeNet-R2N2.** We retrain and re-evaluate our method using different feature conditioning methods.

**Occlusions.** In Tab. 3, we evaluate the performance of CCD-3DR with respect to different occlusion ratios on Pix3D. We randomly mask approximately $20\%$ and $50\%$ visible pixels of the object to test the robustness of CCD-3DR towards occlusions. From the table, it can be seen clearly that although the masked pixels increase from $20\%$ to $50\%$, the performance of CCD-3DR only degrades very little, with 0.013 in *chair*, 0.015 in *table* and 0.024 in *sofa*. Moreover, in this experiment, PC$^2$ also demonstrates consistent and satisfactory results under different occlusion ratios, which verifies the capability of diffusion models in handling occlusions. Note that for fair comparisons, we retrain PC$^2$ and our method with the same augmented training data. We randomly mask $0\% \sim 50\%$ pixels of each image for training and then conduct the ablation study in Tab. 3.

**Local *vs.* Global Conditioning.** In Tab. 4, we demonstrate the effect of local and global features in the diffusion-based reconstruction process. The global feature is obtained by averaging the pooling of the point-wise local features. And when the global feature is incorporated, we directly concatenate it to each point as the condition. Comparing *Ours-(G+L)* and *Ours*, it can be seen clearly that once a local feature is provided, an additional global feature is not necessary.

**Oracle Results.** We report the oracle experiment results in Tab. 1. Following the setting in PC$^2$, we also predict 5 possible shapes for each image and select the one with the highest F-Score@0.01 as the final reconstruction result. It is obvious that under the oracle setting, all three diffusion-based methods, DMPGen*, PC$^2$, and Ours, showcase a significant leap forward. Thereby, although the centralization scheme in our method may influence the generalization capability of the diffusion model to a certain extent, in the single-view reconstruction case, our method still demonstrates the capability of generating multiple plausible results. We also provide the corresponding qualitative results in the Supplementary Material.

## 5 CONCLUSIONS

In this paper, we propose CCD-3DR, a single-image 3D reconstruction pipeline that leverages a novel Centered Diffusion Probabilistic Model (CDPM) for consistent and stable local feature conditioning. We project the predicted noise and sampled point cloud from DDPM into a subspace where the point cloud center remains unchanged during the whole diffusion and reverse processes. Extensive experimental results and ablation studies on both synthetic and real-world datasets demonstrate that such a simple design significantly improves overall performance. We also analyze the influence of point cloud centralization with respect to diversity and point out the limitations of CCD-3DR. In the future, we plan to extend CCD-3DR with an advanced ordinary differentiable equation solver to enhance the inference speed.

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
