# SUPPLEMENTARY MATERIAL OF CCD-3DR: CONSISTENT CONDITIONING IN DIFFUSION FOR SINGLE-IMAGE 3D RECONSTRUCTION

## 1 ADDITIONAL QUALITATIVE RESULTS

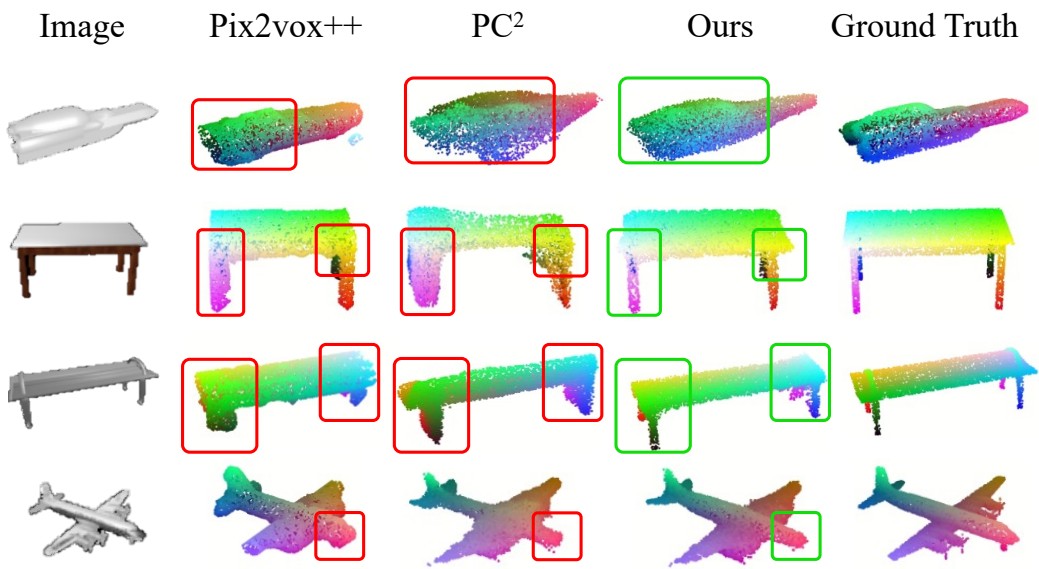

Figure 1: **Additional Qualitative comparisons on synthetic dataset ShapeNet-R2N2**. Rectangles highlight our advantages.

We provide some additional qualitative comparisons on synthetic ShapeNet-R2N2 and real-world Pix3D, as Fig. 1 and Fig. 2 show, respectively. The visualization results demonstrate that CCD-3DR achieves more accurate geometric reconstruction. Compared to diffusion-based baseline PC$^2$, the reconstruction point clouds of CCD-3DR match the input images more closely since the proposed CDPM technique. On the contrary, PC$^2$ presents inferior reconstruction results containing more outliers with significant drifting (the plane of the table in the second row of Fig. 1). Moreover, compared with voxel-based Pix2Vox++, we obtain superior performance when handling tiny structures like the head of the plane in the last row of Fig. 1, since the voxel-based method typically suffers from resolution limitation.

## 2 VISUALIZATION OF THE ORACLE SETTING

Fig. 3 provides the qualitative results of the Oracle setting of our method. It illustrates that in the single-view reconstruction case, our method demonstrates the capability of generating multiple plausible results since it is categorized as a generative model. CCD-3DR maintains the ability to yield multiple reconstructions that tightly match the ground truth on the view angle of the input while completing other details. From the perspective of single-view reconstruction, the capability to match the origin view decides reconstruction accuracy. CCD-3DR achieves superiority by limiting

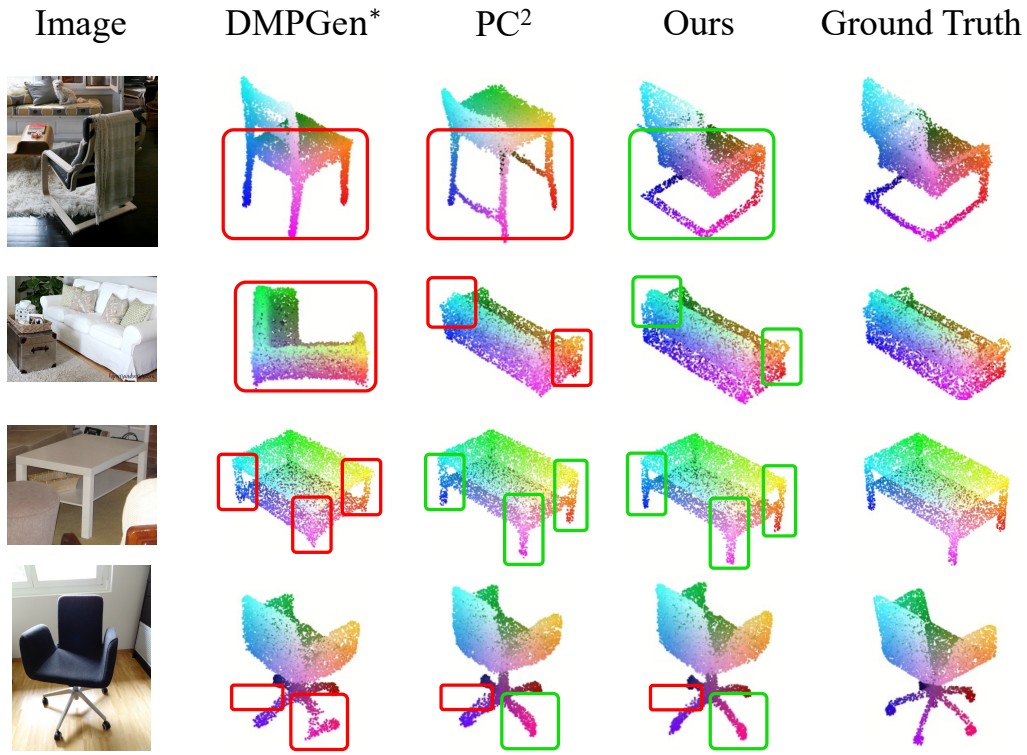

Figure 2: **Additional Qualitative comparisons on real-world dataset Pix3D**. Rectangles highlight the differences.

the sample space for the diffusion model. Meanwhile, the oracle experiment still shows that it holds the generation diversity.

## 3 LIMITATIONS

The biggest limitation is that the generation diversity is less than DDPM theoretically, as CDPM shrinks the sample space of $x^t, \epsilon, \epsilon_\theta$. However, the task of single-image reconstruction focuses on accuracy more than diversity, which aligns with the purpose of CDPM. Another limitation is that when the object is highly self-occluded, the reconstructed results tend to be less satisfying. This seemingly discouraging outcome may be expected, as no such local image feature can be projected and conditioned on the diffusion process. For example, in the last row of Figure. 2, the image does not show the fifth leg of the armchair, so all three methods cannot generate it. Nonetheless, our generation result still tends to be sensible regarding the chair's function (enough standing stability using four legs) and the better generation quality compared to DMPGen.

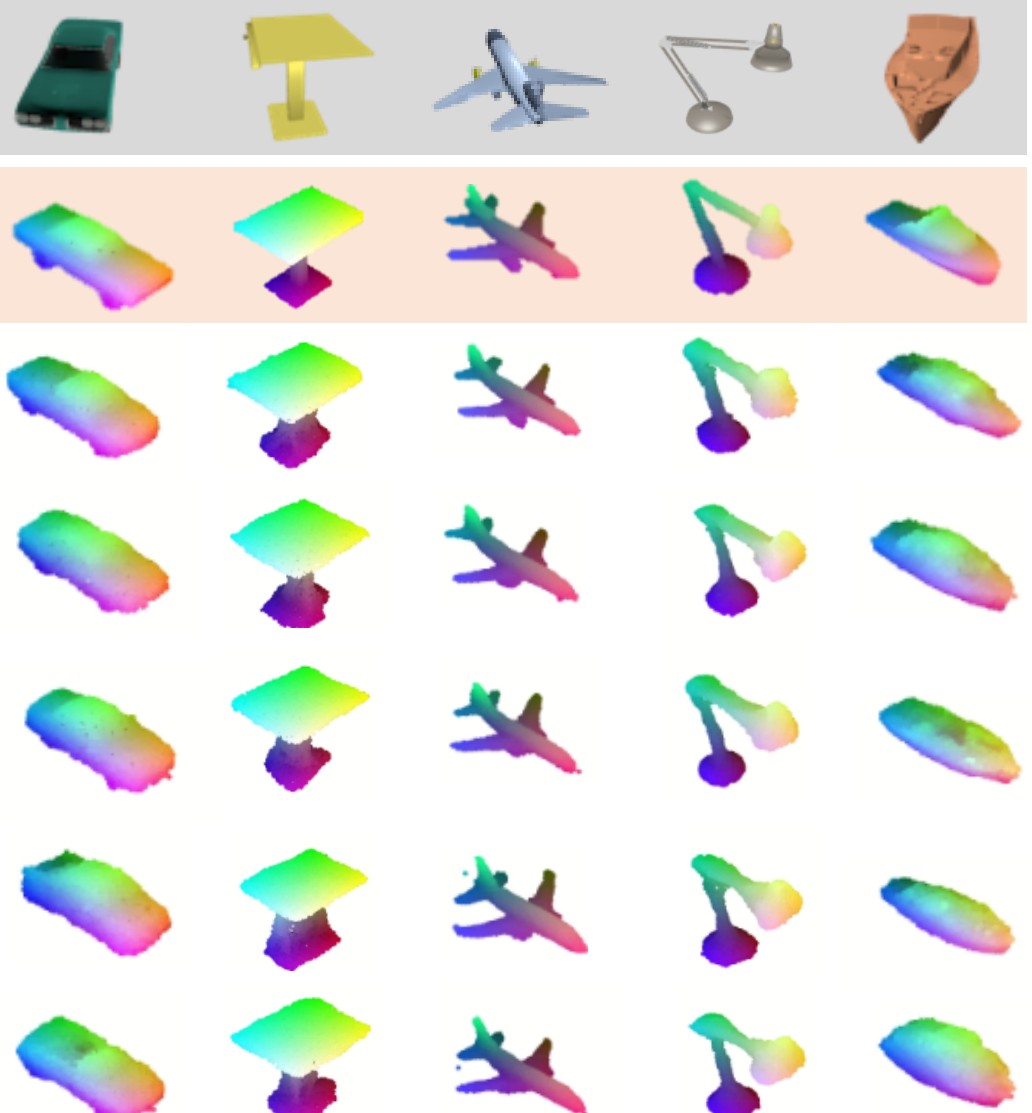

Figure 3: **Visualization of our Oracle results.** The input images and corresponding ground truth shapes are highlighted with grey and orange colors, respectively. The five reconstruction results are sorted according to the F-Score@0.01 w.r.t the ground truth.