# OpenReview forum: "CCD-3DR: Consistent Conditioning in Diffusion for Single-Image 3D Reconstruction"
_ICLR.cc/2024/Conference — ICLR 2024 Conference Withdrawn Submission_

### Official Review · Reviewer_a25m · 2023-10-23

**Soundness:** 2 fair
**Presentation:** 2 fair
**Contribution:** 3 good
**Rating:** 3
**Confidence:** 2

**Summary:**

The paper proposes a method for image-to-3D point cloud prediction. Building on the recent PC^2 method, which conditions the generated point cloud on the image by projecting it onto the image features using known camera parameters, the current method finds it advantageous to explicitly enforce an inductive bias that centers the generated point cloud at each step of denoising. Significant improvements are demonstrated on ShapeNet R2-N2 and Pix3D.

**Strengths:**

- The performance improvement on ShapeNet-R2N2 is impressive. I appreciate that the authors included the oracle evaluation proposed in PC^2. On one hand, this emphasizes the variance in the reconstructions; on the other hand, I believe it makes the comparison more fair.
- The reconstruction improvement over PC^2 and DMPGen on the Pix3D dataset is especially exciting, given that this involves real data.
- The implementation of the centering concept is a straightforward addition to the previous method, PC^2, which is appealing and doesn't incur additional computational cost.

**Weaknesses:**

- Main concern: While the improvement in results is clear and the implementation is simple, I'm currently not convinced by the argumentation. My concern is that the authors propose adding an explicit inductive bias, which assumes that all target models are zero-centered. This assumption may or may not hold for general 3D point cloud generation, depending on how the data is provided. For example, would this be a useful cue if the objects are articulated? In such cases, a folding back action could cause the legs to move, affecting the centering. I'm not convinced by the authors' analysis, motivation, and interpretation. It's crucial that if the paper is accepted based on its strong results, it should also come with a solid understanding of why the results are improving and whether the conditions for improvement will be applicable in other setups.
- Centering in Partial Shapes and Scenes: While centering can serve as an effective canonicalization for complete shapes, it may not be well-defined for partial shapes or scenes. The authors should address this issue. As an aside—while the authors present an experiment with up to 50% missing points, my understanding is that this is merely an augmentation and the ground-truth center is provided. However, this will not be the case in general when training with partial shapes.
- Test-Time Centering: The suggested method modifies both the training and testing processes, suggesting that the improvement is due to better utilization of network capacity. I wonder if this zero-centering could be applied solely as a test-time inductive bias instead? I would be very interested in seeing a comparison that includes this experiment, as it could help validate whether the improvement is indeed due to increased network capacity or the strong inductive bias.
- Clarity: The introduction states that there is a critical problem with the center point shifting during the denoising process. The first issue raised discusses the "wasted" capacity needed to map the center of the Gaussian noise to the center of the final shape. However, the authors do not explain why this mapping is necessary. The assertion that "It is crucial to ensure that the transition of the point cloud center from the initial Gaussian noise state to the final object reconstruction is appropriately managed" requires justification.
- In Section 3.2, the authors open with the statement, "Predicting the center bias is challenging for the network in the reverse process." Why is this the case? What makes it more challenging than any other network prediction? Again, I find the explanations in this section to be somewhat "hand-wavy." Some of the claims appear to require the explicit assumption that each target output is zero-centered—an assumption that is neither guaranteed nor necessarily desirable. Is there any relevance to images in this discussion? I'm not very familiar with the details of 2D diffusion, but I believe there is normalization of the output distribution. However, I don't think the output of each image is assumed to be centered.
- Center of Mass: The authors opted to canonicalize the shapes using the point mean. In cases where the density is non-uniform, this approach could lead to issues. Specifically, very similar shapes that are sampled differently might end up having different "centroids." Did the authors consider other forms of canonicalization, such as using the bounding box?
- Loss of Diversity: The authors mention that there is a sacrifice of diversity in the outputs. This claim needs to be both demonstrated and measured.
- Qualitative results on CO-3D would be nice to see. This data was used in the main baseline work PC^2.
- I’m missing comparison with a NeRF-based methods, like the recent Zero-1-to-3
- I also recommend comparison with point-e
- I don’t see the relevance of the occlusion experiment — it doesn’t seem like the method is proposing anything specific to occlusion.

Minor:

- several statements were unclear to me and would benefit further explanation. like: “each point inside the point cloud and predicted noise is independently modeled” — aren’t the points predicted jointly?

**Questions:**

See weaknesses where I listed my main concern which i hope the authors can address and provide clarity.

---

### Official Review · Reviewer_wYQz · 2023-10-29

**Soundness:** 3 good
**Presentation:** 3 good
**Contribution:** 2 fair
**Rating:** 3
**Confidence:** 4

**Summary:**

Single-image 3D shape reconstruction is a well-known but ill-posed problem, and how to extract stable and informative image features is the key for 3D generation. This paper identifies a neglected issue existed in previous single-image shape reconstruction approaches, especially for DDPM-based approaches, i.e., the center of generated shapes at each time step fluctuated, leading to unstable 2D image feature extraction. To resolve this issue, this work proposed to centerize both noise, diffused point clouds, as well as denoised point couds, to make their centroid stable.

The experiments are conducted on a synthetic dataset ShapeNet-R2N2 and a real-world dataset Pix3D. Quantitative results show that the proposed approach achieves the best performance against the compared methods. Ablation study further validates the robustness of the proposed approach towards occlusions.

**Strengths:**

1. $\textbf{Problem Formulation}$: The strength of the paper is the motivation as it identified a neglected issue in previous approaches, i.e.,  the centroid of the generated shapes shift, leading to inconsistent 2D image feature projection and extraction. This is a good catch and is insightful.

2. $\textbf{Method Soundness}$: Though the method proposed in this work is relatively simple, it is correct and sound. It indeed resolve the identified issue.

3. $\textbf{Experimental Results:}$ By stabilizing the centroid of the denoised shapes in the reverse process of DDPM, it achieves the best performance over the compared approaches, either the one using global image features or the one using local image features through point projection.

**Weaknesses:**

1. $\textbf{Method}$

1.1 The method is indeed very simple, just centerizing everything, including noise, noisy point clouds and denoised point clouds. Though it is effective, the method itself lacks insight and broader impact.

1.2 The subspace part seems to be not necessary. Actually, it is hard to claim these are subspaces in a rigorous mathematical perspective.

2. $\textbf{Novelty}$

Though the performance of the proposed approach is very good, the method itself is somehow more like an effective trick, making it less inspiring and enlightening.

**Questions:**

Please refer to the Weaknesses part.

---

### Official Review · Reviewer_9Ca5 · 2023-10-31

**Soundness:** 4 excellent
**Presentation:** 3 good
**Contribution:** 2 fair
**Rating:** 6
**Confidence:** 3

**Summary:**

This paper presents a method by which DDPM can be applied to sparse points clouds for single image 3D reconstruction.  In particular, the authors focus on how the system can be improved if data were centered at each stage in the pipeline.  Evaluations on real and synthetic data are provided.

**Strengths:**

Method is relatively straight forward and easy to understand.  Seems reproducible.

Evaluations demonstrate unambiguous benefit of the method.  Seems convincing that we should just center our data for this application.

**Weaknesses:**

This submission seems kind of padded with unnecessary components to get it to 9 pages.  Sec 3.3 and Algos 1 and 2 amount to a page long description of how data was centered.

It seems like a relatively minor change.  I feel like this sort of modification would do well if there were a very strong theoretical analysis in the paper, if it were combined with one or two more tricks for the application domain, or if it were applicable to a very broad set of techniques.  I suspect Sec 3.3 was intended to be the theoretical component, but it's pretty weak and states the obvious.

**Questions:**

I don't really have additional questions.  My concerns are not so much the execution of what's in the paper.  It's the scope of what the authors are offering.  If there was a stronger theoretical analysis to take the place of part of Sec 3, maybe that would be persuasive.

---

### Official Review · Reviewer_yrBn · 2023-11-01

**Soundness:** 2 fair
**Presentation:** 3 good
**Contribution:** 2 fair
**Rating:** 5
**Confidence:** 4

**Summary:**

The authors proposed a pipeline for single image to point cloud reconstruction. A Centered Diffusion Probabilistic Model (CDPM) is proposed to solve the point cloud shift/inconsistency problem in the iterative denoising procedure. The authors demonstrate in several small datasets that the proposed method outperforms previous baselines in the task of image to point cloud reconstruction.

**Strengths:**

- The pipeline of the method is clear and easy to understand. Each component looks reasonable.

- Ablation studies have been provided to justify the effectiveness of each component.

- The proposed method outperforms the mentioned baselines. (I did not follow the most recent papers in this field and I will resort to opinions from other reviewers).

**Weaknesses:**

- After the point cloud is centered, it is straight forward that we should change the camera projection matrix correspondingly. However, it looks like the input pose is re-used at each iteration. Why the projection is still valid in such a case?

- Will the point cloud occlusion affect the local image feature extraction?

- The proposed method is only trained and evaluated in small-scale toy datasets and it is hard to access its scalability and applicability in wider scenes.

- The word "back-project" is misused. It should be we can "project" a point to the image, and "back-project" a pixel from the image to a 3D point/viewing ray.

- I would recommend the authors change (first, second, third) to 1) 2) 3) or itemize.

**Questions:**

See above.